# (*E*)-2,6,10-Trimethyldodec-8-en-2-ol: An Undescribed Sesquiterpenoid from Copaiba Oil

**DOI:** 10.3390/molecules26154456

**Published:** 2021-07-23

**Authors:** Mohammed F. Hawwal, Zulfiqar Ali, Mei Wang, Jianping Zhao, Joseph Lee, Omer I. Fantoukh, Ikhlas A. Khan

**Affiliations:** 1Department of BioMolecular Sciences, Division of Pharmacognosy, School of Pharmacy, The University of Mississippi, University, MS 38677, USA; mhawwal@ksu.edu.sa; 2Department of Pharmacognosy, College of Pharmacy, King Saud University, Riyadh 4545, Saudi Arabia; ofantoukh@ksu.edu.sa; 3National Center for Natural Products Research, School of Pharmacy, The University of Mississippi, University, MS 38677, USA; jianping@olemiss.edu (J.Z.); jclee1@olemiss.edu (J.L.); 4Natural Products Utilization Research Unit, Agricultural Research Service, U.S. Department of Agriculture, University, MS 38677, USA; meiwang@olemiss.edu

**Keywords:** copaiba oil, *Copaifera*, Fabaceae, new sesquiterpenoid, sesquiterpenes, NMR

## Abstract

The use of copaiba oil has been reported since the 16th century in Amazon traditional medicine, especially as an anti-inflammatory ingredient and for wound healing. The use of copaiba oil continues today, and it is sold in various parts of the world, including the United States. Copaiba oil contains mainly sesquiterpenes, bioactive compounds that are popular for their positive effect on human health. As part of our ongoing research endeavors to identify the chemical constituents of broadly consumed herbal supplements or their adulterants, copaiba oil was investigated. In this regard, copaiba oil was subjected to repeated silica gel column chromatography to purify the compounds. As a result, one new and seven known sesquiterpenes/sesquiterpenoids were isolated and identified from the copaiba oil. The new compound was elucidated as (*E*)-2,6,10-trimethyldodec-8-en-2-ol. Structure elucidation was achieved by 1D- and 2D NMR and GC/Q-ToF mass spectral data analyses. The isolated chemical constituents in this study could be used as chemical markers to evaluate the safety or quality of copaiba oil.

## 1. Introduction

European settlers reported that the people of the Brazilian Amazon region used copaiba trees to treat their wounds [1]. Marcgraf and Piso were the first to describe the copaiba tree in 1638 [2]. Amazonians use copaiba oil resin as an anti-inflammatory and healing agent [3]. Copaiba oil is obtained by tapping the trunk of the *Copaifera* tree. Copaiba oil can be obtained from several species of *Copaifera* trees, but *Copaifera reticulata* is responsible for 70% of oil production [4,5]. *Copaifera* trees belong to the family Fabaceae (Leguminosae) [4]. *Copaifera* trees range from 0.5 to 5 m in diameter and up to 60 m in height, and can live up to 400 years [4]. Copaiba oil is an exudate secretion that results from the trees’ detoxification process. The secretion acts as a defensive mechanism against ordinary predators, such as fungi and bacteria [6]. South America, particularly Brazil, has diverse *Copaifera* species and is considered the largest global exporter of copaiba oil.

Copaiba oil has been used in folk medicine for centuries as a wound-healing agent [7]. This use was likely inspired by animals, as the indigenous people observed injured animals rubbing their bodies on the stem of *Copaifera* trees [4]. The indigenous people of the Amazon region also used the *Copaifera* tree to treat several other conditions, such as urinary tract infections, sore throats, stomach ulcers, and infectious diseases. *Copaifera* trees play a vital role as an alternative remedy in the Amazon region of Brazil, and it is no surprise that phytotherapeutic and cosmetic products using copaiba oil have found their way not only into the Brazilian market but also to the international markets [1]. The main chemical constituent of copaiba oil is β-caryophyllene, which can be found in various essential oils [8]. The chemical profile of copaiba oil might be slightly different from one species to another, but in general, the main constituents are β-caryophyllene, α-humulene, α-copaene, α-bergamotene, δ-cadiene, and β-bisabolol. Some copaiba oils contain diterpene acids, such as copalic acid, clorechinic acid, and hardwickiic acid [4]. However, several factors might cause chemical composition variation in copaiba oil, such as seasonal and climatic characteristics of the environment, soil type, and composition, rainfall index, and species genetics [4,5]. In this study, copaiba oil was investigated as part of a continuing effort to isolate and identify chemical markers for use in quality studies of dietary supplements and botanical drug products under development in the United States. In this regard, eight sesquiterpenes/sesquiterpenoids (Figure 1), including one new, were isolated and characterized by analyzing their 1D and 2D NMR and GC/Q-ToF mass data.

## 2. Results and Discussion

Copaiba oil was fractionated via silica gel column chromatography into three fractions. Compounds **2–4** were obtained from fraction A and **5–8** were purified from fraction B. Due to the limited amount and complexity of fraction C, only one compound (**1**) could be isolated from it. Compound **1** was obtained as a colorless oil. Its molecular formula, C_15_H_30_O, was established based on the [M-H_2_O]^+^ peak at *m/z* 208.2188 (calcd for C_15_H_28,_
*m/z* 208.2191) in the GC/Q-ToF MS. The IR spectrum of **1** showed characteristic bands for alkene (1709 cm^−1^) and hydroxyl groups (3400 cm^−1^). The ^13^C NMR spectrum of **1** revealed 15 signals corresponded to five methyl groups, two olefinic methines, two aliphatic methines, five methylenes, and an oxygenated non-protonated carbon. The ^1^H NMR spectrum of **1** displayed a singlet of six protons at 1.21 ppm assignable to two tertiary methyl groups (H_3_-1 and H_3_-13), a triplet (*J* = 7.4 Hz) of three protons at 0.84 ppm typical for a primary methyl group (H_3_-12), and two characteristic doublets of three protons each (*J* = 6.6 Hz and *J* = 6.8 Hz) at 0.86 ppm and 0.95 ppm for secondary methyl groups (H_3_-14 and H_3_-15). A *trans* double inferred from two typical olefin resonances at *δ*_H_ 5.24 (dd, *J* = 15.4, 7.5 Hz) and 5.32 (dt, *J* = 15.4, 7.5). The proton resonances were connected to their respective carbons with the help of HSQC spectrum (Table 1). The locations of the hydroxy and two tertiary methyl groups at C-2 were supported by the HMBC correlations of methyl protons (*δ*_H_ 1.20) with C-2 (*δ*_C_ 71.23) and C-3 (*δ*_C_ 44.41). Similarly, the positions of other methyl groups, double bond, methylenes, and methines were confirmed by the HMBC correlations and ^1^H-^1^H COSY couplings (Figure 2). Due to the acyclic nature of the molecule, the absolute configurations at C-6 and C-10 could not be determined. The ^1^H and ^13^C NMR data assignments were carried out by COSY, HSQC, and HMBC NMR data analyses, and compound **1** was ultimately elucidated as a sesquiterpenoid named (*E*)-2, 6, 10-trimethyldodec-8-en-2-ol. To the best of our knowledge, compound **1** was found to be a new molecule based on a SciFinder database search.

Previously described compounds (Figure 1) were identified via their 1D and 2D NMR and mass data analyses, which were in agreement with those reported in the literature, as β-caryophyllene (**2**) [9], α-humulene (**3**) [10], caryophyllene oxide (**4**) [11], malliol (**5**) [12], T-cadinol (**6**) [13], T-muurolol (**7**) [14] and torreyol (**8**) [15]. GC/MS analysis of the investigated copaiba oil showed the following percent composition of the aforementioned compounds: 0.01% (*E*)-2,6,10-trimethyldodec-8-en-2-ol (**1**), 38.58% β-caryophyllene (**2**), 6.26% α-humulene (**3**), 0.02% caryophyllene oxide (**4**), 0.04% malliol (**5**), 0.12% T-cadinol (**6**), 0.16% T-muurolol (**7**), and 0.12% torreyol (**8**) (Appendix A). Quantitative analysis was determined as the peak area percentage utilizing the total ion chromatogram (TIC).

Several biological activities have been reported for the isolated compounds. β-caryophyllene (2) has been associated with numerous biological activities, such as anticarcinogenic, antibiotic, and anti-inflammatory [16]. Anti-inflammatory properties of α-humulene (**3**) have also been evaluated [17], revealing that oral administration of α-humulene (**3**) produces significant inhibitory effects in multiple mouse and rat models. Caryophyllene oxide (**4**) exhibited anti-inflammatory, analgesic [18], and antifungal activities [19]. Essential oil from *P. cylindraceus* containg 42% maaliol (**5**) demonstrated significant cytotoxic activity against three cancerous cell lines (Hela, HepG2, and HT-29) [20]. T-cadinol (**6**) produced smooth-muscle relaxant effects on isolated guinea pig ileum, depending on the dosage [21]. T-Muurolol (**7**) showed remarkable antifungal and antitermitic activity [22]. Torreyol (**8**) revealed antiproliferative and cytotoxic activities [23].

## 3. Materials and Methods

### 3.1. General Procedures

NMR spectra were measured on a Bruker AU III 500 MHz NMR spectrometer (Bruker, Billerica, MA, USA) and chemical shifts were referenced to the residual solvent signals. IR spectrum was carried out on Agilent Technologies Cary 630 FTIR (Agilent Technologies, Santa Clara, CA, USA). GC/Q-ToF mass data were recorded on an Agilent 7890 GC and the Agilent 7250 Accurate-Mass Quadrupole Time-of-Flight mass spectrometer equipped with an electron ionization source operated with an electron energy of 70 eV. The data were acquired utilizing Agilent MassHunter software (version B7.06.274). Flash silica gel (32–63µ, Dynamic Adsorbents Inc, Norcross, GA, USA) was used as an adsorbent in column chromatography (CC). TLC was performed on silica gel F_254_ aluminum sheet (20 × 20 cm) (Sorbent Tech., Norcross, GA, USA). Spots were visualized on TLC by spraying with 0.5% vanillin (Sigma, St. Louis, MO, USA) solution in conc. H_2_SO_4_–EtOH (5:95) followed by heating. Analytical grade solvents (Fisher Chemicals, Hampton, NH, USA) were used for purification.

### 3.2. Plant Material

The copaiba oil obtained from *Copaifera officinalis* L. was provided by doTERRA (Pleasant Grove, UT, USA). A reference sample # 613 was deposited in the product repository at the National Center for Natural Products Research (NCNPR), University of Mississippi.

### 3.3. Extraction and Isolation

The oil (13.3 g) was subjected to column chromatography (CC) over silica gel (105 cm × 5 cm) using a stepwise gradient concentration of solvents system starting with 100% hexanes (3 L) followed by increasing polarity with ethyl acetate till 30% ethyl acetate was reached (10% ethyl acetate (1.5 L), 10% ethyl acetate (1.5 L), and 10% ethyl acetate (1 L)). Based on TLC profile, the resulting fractions were merged into three collective fractions (A, B, and C). Fraction A (8.5 g) was applied to CC over silica gel (105 cm × 4 cm) using hexanes/ethyl acetate (100:0 (1.5 L), 97:3 (1 L), and 95:5 (1 L)) to yield 23 fractions (A1-A23). β-Caryophyllene (**2**, 700 mg) from Fr. A6, α-humulene (**3**, 12 mg) from Fr. A10, and caryophyllene oxide (**4**, 13.5 mg) from Fr. A15 were purified by CC over silica gel (85 cm × 3 cm) using mixtures of hexanes/ethyl acetate (98:2 (1.5 L), 97:3 (1.5 L), and 90:10 (1.5 L), respectively). Fraction B (2 g) was chromatographed over silica gel (75 cm × 3 cm) using a mixture of hexanes/ethyl acetate (90:10 (2 L)) to yield six fractions (B1-B6). T-Cadinol (**6**, 10.5 mg) from Fr. B1, T-muurolol (**7**, 14.4 mg) from Fr. B2, and torreyol (**8**, 9.4 mg) from Fr. B3 were obtained by CC over silica gel (75 cm × 2 cm) using a mixture of hexanes/ethyl acetate (90:10, (500 mL)). Fraction B4 was subjected to CC on silica gel (65 cm × 2 cm) using hexanes/ethyl acetate (80:20 (400 mL)) mixture to yield maaliol (**5**, 2.5 mg). Compound **1** (2.9 mg) was obtained from Fr. C (215 mg) by CC over silica gel (72 cm × 2 cm) with hexanes/ethyl acetate (70:30 (500 mL)) mixture.

### 3.4. Spectral Data

***(E)-2, 6, 10-trimethyldodec-8-en-2-ol*** (**1**). Colorless oil. [α]^23^_D_ 66 (*c* 0.16, CHCl_3_). IR ν_max_ cm^−1^: 3400, 2924, 2855, 1709, 1469, 1377, 1153, 971, 907. NMR data: see Table 1. GC/Q-ToF-MS: *m/z* 208.2188 [M-H_2_O]^+^ (calcd for C_15_H_28,_
*m/z* 208.2191), 179.1794 [C_13_H_23_]^+^, 123.1168 [C_9_H_15_]^+^, 109.1012 [C_8_H_13_]^+^, 95.0855 [C_7_H_11_]^+^, 81.0699 [C_6_H_9_]^+^, 69.0699 [C_5_H_9_]^+^, 67.0542 [C_5_H_7_]^+^, 55.0542 [C_4_H_7_]^+^, *m/z* 41.0386 [C_3_H_5_]^+^.

### 3.5. GC/MS Analysis

For a general analysis of copaiba samples, an Agilent 7890 GC equipped with a 7693 autosampler and an Agilent 5975C quadrupole mass spectrometer was used. Separation was achieved with an Agilent DB-5MS Ultra Inert column (60 m × 0.25 mm × 0.25 µm). The inlet was held at 260 °C and was operated in the split mode with a split ratio of 50:1 and an injection volume of 1uL. The initial GC oven temperature was 80 °C; it was then heated at 3 °C/min to 125 °C, then ramped at 1 °C/min to 140 °C and held for 10 min, then heated at 3 °C/min to 170 °C, before a final ramp of 8 °C/min to 280 °C held for 10 min was used. The mass spectrometer was equipped with an electron ionization source, which was operated with an electron energy of 70 eV. The ion source, quadrupole, and transfer line temperatures were set to 230, 150, and 280 °C. Mass spectra data were recorded from 35 to 500 *m/z* after a 5 min solvent delay. Data were acquired utilizing Agilent MassHunter software (version B7.06.274). All of the copaiba samples were diluted in dichloromethane (0.01%, *v*/*v*), and n-dodecane (IS) was added to each sample solution at a constant concentration of 300 µg/mL.

In order to obtain the accurate mass of compound 1, analysis was performed utilizing an Agilent 7890B gas chromatographic (GC) instrument which was equipped with a RS185 PAL3 autosampler. The GC was connected to an Agilent 7250 Accurate-Mass Quadrupole Time-of-Flight (Q-ToF) mass spectrometer. The capillary column (30 m × 0.25 mm i.d.) utilized was coated with a 5% Phenyl Methyl Siloxane (J&W HP-5MS) film (0.25 µm). Helium at a constant flow rate of 1 mL/min was used as the carrier gas. The following GC oven program was utilized: 50 °C held for 1 min and then heated at a rate of 6 °C/min to 260 °C. The inlet was programmed at 260 °C in split mode. A split ratio of 50:1 with an injection volume of 0.2 uL was used for compound **1**. The Q-ToF mass spectrometer was equipped with a high-emission low-energy electron ionization source which was operated with an electron energy of 70 eV and an emission current of 5.0 µA. The source, quadrupole, and transfer line temperatures were 230 °C, 150 °C, and 260 °C, respectively, during the experiment. All mass spectra data were recorded at a rate of 5 Hz from 35 to 450 m/z after a 4 min solvent delay. Data were acquired utilizing Agilent MassHunter software (version B7.06.274).

## 4. Conclusions

In this study, eight compounds were isolated from the copaiba oil. All compounds were characterized as sesquiterpenes/sesquiterpenoids. Out of them, (*E*)-2,6,10-trimethyldodec-8-en-2-ol (**1**) was found to be previously undescribed. Possible contribution of the new compound (**1**) to the biological activity of copaiba oil will have to be investigated in further studies.

## Figures and Tables

**Figure 1 molecules-26-04456-f001:**
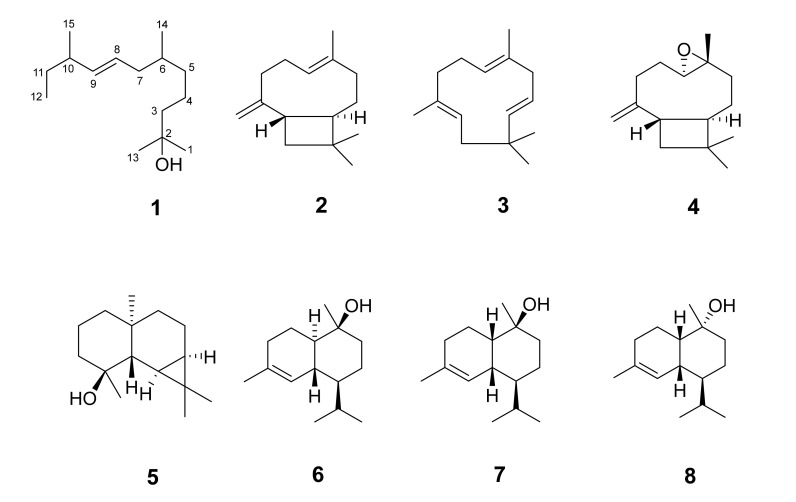
Chemical structures of the isolated compounds.

**Figure 2 molecules-26-04456-f002:**
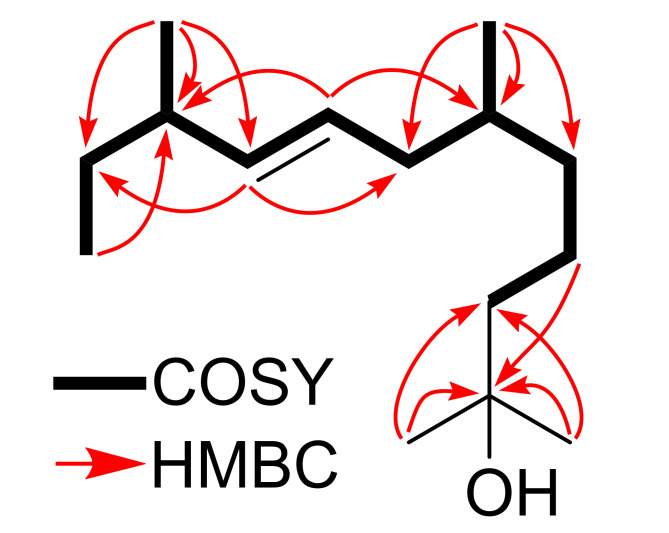
COSY couplings and key HMBC correlations of compound **1**.

**Table 1 molecules-26-04456-t001:** ^13^C NMR (125 MHz) and ^1^H NMR (500 MHz) spectral data of **1** in CDCl_3_.

No	δ_C_	δ_H_, mult. (*J* in Hz)
1	29.37	1.21, s
2	71.23	-
3	44.41	1.40, m 1.45, m
4	21.96	1.32, m 1.38, m
5	37.19	1.11, m 1.31, m
6	33.34	1.45, m
7	40.23	1.82, dt (14.0, 7.5) 2.00, dt (14.0, 7.5)
8	127.10	5.32, dt (15.4, 7.5)
9 10	137.74 38.69	5.24, dd (15.4, 7.5) 1.96, m
11	30.05	1.26, m 1.29, m
12	11.97	0.84, t (7.4)
13	29.42	1.21, s
14	19.62	0.86, d (6.6)
15	20.73	0.95, d (6.8)

## Data Availability

Not applicable.

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
