# Peer review of "(E)-2,6,10-Trimethyldodec-8-en-2-ol: An Undescribed Sesquiterpenoid from Copaiba Oil"

_molecules, 2021, doi:10.3390/molecules26154456_

Round 1

Reviewer 1 Report

The presented work provides information on metabolites isolated from Copaiba oil. The work provides data on the establishment of the structure of an only one new compound related to acyclic sesquiterpenoid.  In my opinion, the work is not of great scientific interest. The structural formula of the new substance is very simple (most likely it is a biogenetic precursor of other compounds identified by the authors), it has been established correctly. However, no attempts have been made to describe even the relative configuration of asymmetric centers (NOESY / ROESY experiments have not been removed). There is no data on the biological activity of the isolated compounds. The work is well-framed, but it is done at a low methodological level and does not represent scientific value.

I think that in terms of the level of performance and the quality of the material provided, the work cannot be published in the Molecules journal.

Author Response

Response.

As the new compound does not have cyclic unit, the NOESY / ROESY experiments will not help to determine the relative configuration. The reported biological activities of the known compounds have been included as per the suggestions from the other reviewers.

Reviewer 2 Report

This manuscript reports on the isolation of a new compound and seven known compounds from copaiba oil. This manuscript presents valuable research, however, under this reviewer’s opinion, it needs some improving.

The abstract could be improved. A few lines could be added mentioning separation method and main conclusions. A few more keywords could also be included.

The introduction is appropriate, but nothing was remarked about the chemical composition of copaiba oil. Since it is a research article on the isolation of natural products from copaiba oil, some lines could be added about the chemistry of this oil. According to Molecules template, it is recommended that Fig. 1 is placed right after it was mentioned for the first time (before Section 2).  Please, correct Fig. 1 in order to avoid atoms overlapping in some of the molecules.

Discussion was focused only in the structure elucidation; authors could discuss about the isolation of the compounds reported in this manuscript. Since the introduction focuses a lot on the biological activity of copaiba oil, some remarks could be made about the reported biological activity of the known compounds isolated from fractions A and B.

Regarding the materials and methods, several compounds were isolated from fractions A (8.5 g) and B (2 g), but only compound 1 was isolated from C, were there no other compounds in this fraction? The weight of fraction C was not mentioned. Please, include the integration in the 1H NMR data. No FTIR data was included regarding the new compound. In Table 1, 500 MHz is indicated twice (it was already mentioned for each experiment, 13C and 1H, the last time is not necessary). C10 is missing in the table.

Conclusions could be improved or removed and included in the discussion section.

Minor corrections: trans (in italics), “double bond” (line 59)

Author Response

Reviewer 2.

This manuscript reports on the isolation of a new compound and seven known compounds from copaiba oil. This manuscript presents valuable research, however, under this reviewer’s opinion, it needs some improving.

The abstract could be improved. A few lines could be added mentioning separation method and main conclusions. A few more keywords could also be included.

Response: The abstract has been modified as per suggestion and a couple of keywords have been added.

The introduction is appropriate, but nothing was remarked about the chemical composition of copaiba oil. Since it is a research article on the isolation of natural products from copaiba oil, some lines could be added about the chemistry of this oil. According to Molecules template, it is recommended that Fig. 1 is placed right after it was mentioned for the first time (before Section 2).  Please, correct Fig. 1 in order to avoid atoms overlapping in some of the molecules.

Response: Information regarding the chemistry of copaiba oil has been added as per suggestion. The location of Figure 1 can be adjusted by the production team at the time of production. Atoms overlapping in Fig. 1 has been fixed.

Discussion was focused only in the structure elucidation; authors could discuss about the isolation of the compounds reported in this manuscript. Since the introduction focuses a lot on the biological activity of copaiba oil, some remarks could be made about the reported biological activity of the known compounds isolated from fractions A and B.

Response: Isolation of the compounds is given in the materials and methods section. Some remarks about the reported biological activity of the isolated known compounds have been added as per suggestion.

Regarding the materials and methods, several compounds were isolated from fractions A (8.5 g) and B (2 g), but only compound 1 was isolated from C, were there no other compounds in this fraction? The weight of fraction C was not mentioned. Please, include the integration in the 1H NMR data. No FTIR data was included regarding the new compound. In Table 1, 500 MHz is indicated twice (it was already mentioned for each experiment, 13C and 1H, the last time is not necessary). C10 is missing in the table.

Response: The amount of Fraction C (215 mg), which has been mentioned in the manuscript now, was limited and contained only one prominent compound. The integration has been mentioned in the 1H NMR spectrum. FTIR data was not included as compound 1 does not have any characteristic functional group. In Table 1, duplication of “500 MHz” has been fixed. The 1H- and 13C NMR data of C-10 has been added now in Table 1.

Conclusions could be improved or removed and included in the discussion section.

Response: The conclusion has been extended.

Minor corrections: trans (in italics), “double bond” (line 59)

Response: ‘Trans’ is italics now.

Reviewer 3 Report

The authors described eight natural products inlcuding a new linear sesquiterpene from the copaiba oil. The introduction and structural elucidation look fine. However, several issues need to be considered for manuscript improving. 

  1. The manuscript type ‘Article’ may not be suitable for this manuscript due to the limited context.
  2. The activity of isolates should be indicated by biological tests or discussed through analyzing known biological data in literature.
  3. The GC-MS spectrum was not clear. Did the authors try ESI-MS or APCI-MS?
  4. The proton and carbon NMR data for C-10 should be provided in table 1. 

Author Response

Reviewer 3.

The authors described eight natural products inlcuding a new linear sesquiterpene from the copaiba oil. The introduction and structural elucidation look fine. However, several issues need to be considered for manuscript improving. 

  1. The manuscript type ‘Article’ may not be suitable for this manuscript due to the limited context.

Response: We are flexible regarding the selection of the type of paper.

  1. The activity of isolates should be indicated by biological tests or discussed through analyzing known biological data in literature.

Response: Some remarks about the reported biological activities of the isolated known compounds have been added as per suggestion.

  1. The GC-MS spectrum was not clear. Did the authors try ESI-MS or APCI-MS?

Response: The mass data was acquired multiple times on GC equipped with sophisticated QToFMS (Quadrupole time-of-flight mass spectrometer) and always got water loss ion instead of a parent ion. The ESI-MS failed to generate the mass ion of compound 1 even tried multiple times in positive and negative ions mode.

  1. The proton and carbon NMR data for C-10 should be provided in table 1. 

Response: The 1H- and 13C NMR data of C-10 has been added now in Table 1.

Round 2

Reviewer 1 Report

Dear Authors!
I still believe that the scientific novelty in your material is minimal. Yes, I agree that it is not possible to establish relative and absolute configurations of asymmetric centers by chemical methods. However, the absence of this information significantly diminishes the value of the scientific information you provide. There is no novelty necessary for a scientific publication! You cited data on biological activity, but this is literary data, you are not writing a review.
In my opinion, the presented work cannot be published in this journal.

Author Response

Response: The literature base data was added as per another reviewer’s suggestion. 

Reviewer 2 Report

After carefully revising this manuscript, most of the suggestions have been followed and the abstract, introduction, discussion and conclusion sections were improved.

The bioactivity discussion is well sustained and contains some recent and robust references. However, after the listing of the activity of each compound, a concluding or closing remark could be included discussing the importance of these compounds contained in copaiba oil for its biological importance (as mentioned in the introduction, it is used for folk medicine).

Regarding isolation of the compounds, although it is described in the materials and methods section, this reviewer meant some remarks about the procedure could have been included in the discussion. For example, discussing the yield of the different compounds in the oil or some remarks trying to explain why only one prominent compound was found in fraction C (representing 1.3% of the total weight). Is it because complexity of the fraction C? Too many minor compounds?

FTIR data should be included in the SI, the compound has two characteristic functional groups: a hydroxyl and a trans-double bond, and the ν-OH should be quite prominent in the range of 3500–3200 cm-1. FTIR spectra are unique for each compound, if the chemical composition of copaiba oil might be used for quality control (as mentioned in the manuscript), the FTIR spectra is important, since it is employed nowadays for different applications in industry.

Minor corrections: ‘/’ symbol after sesquiterpenes in the abstract, ‘ad’ (line 54).

Author Response

The bioactivity discussion is well sustained and contains some recent and robust references. However, after the listing of the activity of each compound, a concluding or closing remark could be included discussing the importance of these compounds contained in copaiba oil for its biological importance (as mentioned in the introduction, it is used for folk medicine).

Response: Concluding remark in this regard has been added in the conclusion section.

Regarding isolation of the compounds, although it is described in the materials and methods section, this reviewer meant some remarks about the procedure could have been included in the discussion. For example, discussing the yield of the different compounds in the oil or some remarks trying to explain why only one prominent compound was found in fraction C (representing 1.3% of the total weight). Is it because complexity of the fraction C? Too many minor compounds?

Response: As per kind suggestion, the information has been added immediately after Results and discussion heading and at lines 88-91.

FTIR data should be included in the SI, the compound has two characteristic functional groups: a hydroxyl and a trans-double bond, and the ν-OH should be quite prominent in the range of 3500–3200 cm-1. FTIR spectra are unique for each compound, if the chemical composition of copaiba oil might be used for quality control (as mentioned in the manuscript), the FTIR spectra is important, since it is employed nowadays for different applications in industry.

Response: FTIR spectrum has been provided in the supplementary file and information regarding IR has been added in the Results and discussion as well as in the spectral data sections.

Minor corrections: ‘/’ symbol after sesquiterpenes in the abstract, ‘ad’ (line 54).

Response: Corrected as per suggestion.

Reviewer 3 Report

The authors have modified the manuscript as suggested. It might be accepted. However, there is only one point for further considerison. It is the mass spectrum with abundant MS fragments that looks interesting. If there are no clear mass spectrum indicating the molecular formula of 1, the authors should explain the mass fragment ions, such as 123.1171, 109.1015, 95.0857, 81.0700, 67.0543. The proposed fragment pathway is prefered. 

Author Response

Response: The fragment ions have been added in the spectral data section and the proposed fragment pathway has been provided in the supplementary file (fig.8S).